# Topological Resistance-Free One-Way Transport in a Square-Hexagon Lattice Gyromagnetic Photonic Crystal

**DOI:** 10.3390/nano12173009

**Published:** 2022-08-30

**Authors:** Qiumeng Qin, Jianfeng Chen, Hao Lin, Chaoqun Peng, Zhi-Yuan Li

**Affiliations:** 1School of Physics and Optoelectronics, South China University of Technology, Guangzhou 510640, China; 2State Key Laboratory of Luminescent Materials and Devices, South China University of Technology, Guangzhou 510640, China

**Keywords:** topological one-way edge states, square-hexagon lattice, resistance-free transport, gyromagnetic photonic crystal

## Abstract

We theoretically propose and experimentally realize a new configuration of a photonic Chern topological insulator (PCTI) composed of a two-dimensional square-hexagon lattice gyromagnetic photonic crystal immersed in an external magnetic field. This PCTI possesses five distinct types of edges and all of them allowed the propagation of truly one-way edge states. We proceeded to utilize this special PCTI to design topological transmission lines of various configurations with sharp turns. Although the wave impedances of the edge states on both sides of the intersections in these transmission lines were very different, definitely no back reflection occurred and no mode-mixing problems and impedance-mismatching issues at the intersections were present, leading to topological resistance-free one-way transport in the whole transmission line network. Our results enrich the geometric and physical means and infrastructure to construct one-way transport and bring about novel platforms for developing topology-driven resistance-free photonic devices.

## 1. Introduction

Photonic crystals (PhCs) waveguides are the fundamental and necessary building block for numerous PhCs-based devices, e.g., photonic integrated circuits [1,2], directional couplers [3,4], beam splitters [5,6] and so on. Generally, the lengths of practical PhCs waveguides are finite, so any waveguide in realistic photonic circuits or devices unavoidably terminates at some intersections or discontinuities and tends to transit from single modes to multi-modes locally. Especially, at the intersections or discontinuities, once the distinct waveguide modes mutually transform and mix, the impedance matching of them will become difficult, thereby leading to strong back reflections. This surely is an important problem that cannot be neglected in the design of practical PhCs-based optical devices. Hence, reducing or avoiding mode-mixing problems at intersections and impedance-mismatching issues at discontinuities becomes a crucial issue that must be taken into full account in improving the transport efficiency of PhCs-based circuits and devices [7,8,9]. If one does not have to deal with such mode-mixing problems and impedance-mismatching issues, the design of PhCs-based optical devices will become much simplified.

In a vastly different field of topological photonics, the routes, methods and technologies to control the motion of photons have witnessed fantastic development over the past ten years [10,11,12,13,14,15]. Photonic Chern topological insulators (PCTIs), like their electronic counterparts, are known mostly for the unique properties of the one-way edge states involved in them. A prototypical example is the existence of one-way edge states in a magnetized gyromagnetic photonic crystal (GPC) whose time-reversal symmetry is broken [16,17,18,19,20,21]. In this scenario, one-way edge states emerge with topological protection properties, allowing photons to propagate in one direction and forbidding backscattering under any type of imperfections. Utilizing this one-way property, one can construct numerous useful photonic devices, such as topological dispersionless slow light states [22,23,24], topological splitters [6,25], topological lasers [26] and topological antennas [27,28]. Contextually, a natural question appears: can PCTIs offer an effective solution to avoid mode-mixing problems and impedance-mismatching issues at terminates, intersections and discontinuities in designing PhCs-based circuits and devices, besides the one-way transport property? Besides, the vast majority of well-studied PCTIs have been based on the basic square [17,18,29,30], honeycomb [20,31,32,33] and triangular [34,35,36] lattices, but limited by the simple geometrics of the lattices, these PCTIs only possess relatively fewer edge types. Then, it is interesting to build a PCTI with more edge types for the construction of transmission lines with high design flexibility and modulation freedom, and to see whether they still can provide significant improvement and simplification for energy transport to implement the topological resistance-free transport of electromagnetic (EM) waves.

Here, we construct a new configuration of PCTIs in a two-dimensional microwave GPC of a composite square-hexagon lattice. We calculate the band structure, projected band diagram and Chern number of each bulk band to determine the emergence of one-way edge states. In sharp contrast to the aforementioned PCTIs with the basic lattices, this PCTI has five types of edges and all of them are found to support the existence of a one-way guided mode. We utilize the one-way nature of the edge states to overcome various mode-mixing problems at intersections and impedance-matching issues at discontinuities, and to establish the topological resistance-free transmission lines of various configurations with sharp bends. In these transmission lines, although the wave impedances of the edge states on both sides of discontinuities are extremely distinct, the energy fluxes still can propagate forward along the composite lines in a resistance-free way, in stark contrast to conventional transmission lines where energy fluxes will be reflected back at discontinuities and affect the transport efficiency. These results will enrich the geometric and physical infrastructure for the implementation of PCTIs with one-way edge states and will bring about the novel paths for designing topological resistance-free photonic devices.

## 2. Topological One-Way Edge States at Various Edges

We considered the GPC structure with a square-hexagonal array of yttrium iron garnet (YIG) cylinders (3.0 mm in diameter, 5.0 mm in height), as seen in Figure 1a. The lattice constants were *a* = 14.0 mm and *b* = $14\sqrt{2}$ mm along the *x* and *y* directions, respectively. Note that, unlike the basic square, honeycomb and triangular lattices that only possess relatively fewer edge types, this GPC inherently possessed five types of edges, labeled as Edge 1, 2, 3, 4 and 5 by five colors, respectively, as shown in Figure 1b. The structural and material properties of the GPC can be viewed in the Appendix A. Additionally, the band structures were calculated by using the commercial software COMSOL MULTIPHYSICS with the RF module in the frequency domain [37], and only the E polarization state (where the electric field E was parallel to the *z*-axis direction) was considered (see the Appendix A for the experiment schemes and the calculation methods).

It is known that the Chern number can be utilized to predict the existence of one-way edge states in a GPC [38]. Here, we calculated the Chern number numerically via the Wilson-loop approach [39], in which the Chern number of the *n*th band is defined as $C_{n}=\frac{1}{2\pi }\int_{-\pi }^{\pi }{d\theta }_{n,k_{y}}$, where $\theta_{n,k_{y}}\equiv \int_{-\pi }^{\pi }dk_{x}\Lambda_{n,\vec{k}}^{\left(x\right)}$ is the Berry phase for the *n*th band along the loop of $k_{x}$ for a fixed $k_{y}$, and $\Lambda_{n,\vec{k}}^{\left(x\right)}$ is x component of the Berry connection. Our calculations showed that the Chern number of the first, second, third and fourth bands was zero, while the total Chern number of the intersecting fifth and sixth bands was −1, as depicted in Figure 2. Thus, the sum of the Chern number below the second bandgap (yellow region) was −1, indicating that there existed single-mode one-way edge states in the bandgap. We proceeded to perform theoretical calculations and experimental measurements to verify the existence and transport robustness of the one-way edge states for these five edges, as viewed in the Appendix A.

## 3. Topological Resistance-Free One-Way Transport

Now, we examine the transport behaviors of energy fluxes in a transmission line composed of different types of edges, so as to explore and demonstrate how PCTIs can overcome the mode-mixing problems and impedance-mismatching issues at the terminals. In general, two simple but very useful parameters, the reflection coefficient *r_n,n_*_+1_ = (*Z_n_*_+1_ − *Z_n_*)/(*Z_n_*_+1_ + *Z_n_*) and transmission coefficient *t_n_*_,*n*+1_ = 2*Z_n_*_+1_/(*Z_n_*_+1_ + *Z_n_*), can be used to characterize the degree of impedance mismatching [7,8,9]. As shown in Figure 3, for the *n*th interface, *Z_n_* and *Z_n_*_+1_ are the wave impedances for the input *n*th PhC waveguide and the output (*n* + 1)th PhC waveguide, respectively. The wave impedance is usually proportional to the group velocity of the guided mode (*v_g_* = *d**ω*/*dk_x_*); thus, the equation of the reflection and transmission coefficients can be expressed as *r_n_*_,*n*+1_ = (*v_g,n_*_+1_ − *v_g,n_*)/(*v_g,n_*_+1_ + *v_g,n_*) and *t_n_*_,*n*+1_ = 2*v_g,n_*_+1_/(*v_g,n_*_+1_ + *v_g,n_*). When the optimum coupling of energy into the system is achieved, *r_n_*_,*n*+1_ = 0 and *t_n_*_,*n*+1_ = 1, so if one wants to obtain perfect impedance matching at a terminate, *v_g,n_*_+1_ must be equal to *v_g,n_*. We calculated the group velocities of Edges 1–5 at center frequency *f_s_* = 17.76 GHz to estimate the transmission coefficient at the intersections. The calculations showed that the group velocities of Edges 1–5 were *v_g_*_1_ = 0.213*c*, *v_g_*_2_ = 0.176*c*, *v_g_*_3_ = 0.006*c*, *v_g_*_4_ = 0.299*c* and *v_g_*_5_ = 0.115*c*, respectively. Obviously, the group velocities of the edge states at different types of edges were distinctly different; thus, there must have been some impedance mismatching occurring at the intersections connecting any two types of edges, which might affect the transmission efficiency in an ordinary sense of knowledge in optics. However, this rule was no longer effective to the transmission lines constructed by the PCTI.

We constructed two transmission lines with sharp turns in the square-hexagon lattice PCTI. One configuration is illustrated in Figure 4a, where two metallic strips were placed at the upper Edge 1 and left Edge 4 to form a transmission line with a 90° turn. The microwave absorbers were placed close to the lower Edge 1 and right Edge 4 to avoid the reflection of the EM waves. The simulated wave transport diagram displayed in Figure 4b clearly shows that the energy fluxes could unidirectionally transport along the Edge 1, bypass the 90° turn and continue to transport along Edge 4 in a reflection-free way, despite the sharp bend and serious impedance mismatching between Edge 1 and Edge 4. The transmission parameters S21 and S12 represent the forward and reverse transmission coefficients, respectively. Thus, the measured transmission spectra displayed in Figure 4c further confirm that the EM waves at the frequency range of 17.45~17.95 GHz showed an excellent one-way property when transporting along this complicated transmission line.

We proceeded to add a metallic strip close to the lower Edge 1 to build a more complicated transmission line with two 90° turns, as shown in Figure 4d. The simulated field pattern, as illustrated in Figure 4e, showed that the EM wave perfectly traveled from the upper Edge 1 to the lower Edge 1 in a resistance-free way that seemed to ignore the existence of the two sharp turns and serious impedance-mismatching issues. The experimental measurement data illustrated in Figure 4f showed that there still existed a big contrast (about 35 dB) between the parameters S21 and S12 at 17.45~17.95 GHz, indicating good one-way transport behavior. It is worth noting from Figure 4b,e that there appeared no strong electric field localization at the sharp turns, indicating that no strong partial scattering produced by mode-mixing problems occurred. These results clearly indicate that in these transmission lines, although the wave impedances of the lines on both sides of the intersections differed very much, the energy fluxes still transported in only one direction along the transmission lines with no resistance. These key features of the topological transmission line can be attributed to the topologically protected removal of the back reflections in each transport line, as a result of which the standing waves produced by the back reflections between the neighboring transport lines were forbidden and the smooth propagation of energy fluxes along complicated composite transmission lines was ensured.

We continued to introduce three metallic strips close to the boundaries of the GPC to form an even more complicated composite transmission line with six 90° turns. The first structure is illustrated in Figure 5a for a realistic sample in the experiment, where each 90° turn connected two types of edges (including Edge 1 and Edge 4). Figure 5b illustrates that this transmission line allowed EM wave propagating in one direction and forbade back reflections under sharp turns and impedance-mismatching issues. The measured transmission spectra still remained a strong nonreciprocity dwelling in the yellow region (17.45~17.95 GHz), as shown in Figure 5c. Beyond the robust transport, these results imply the possibility of implementing topological resistance-free transport in a more complex transmission line with various types of edges.

The second transmission line structure is illustrated in Figure 5d, which was a very complicated device consisting of six 90° turns, but now connected all five types of edges. Each of the six 90° turns connected two types of edges with different wave impedances; especially, the wave impedance of Edge 3 was much larger than that of the other edges. We investigated the transport behaviors of the energy fluxes in this extremely complicated multi-bend transmission line both numerically and experimentally. Figure 5e reveals that the energy fluxes could transport leftwards at the upper Edge 1, pass through the six sharp corners (90° turn) and five composite edges one by one smoothly and finally reach the lower Edge 1 with no back scattering, mode-mixing problems and impedance-mismatching issues. The measured transmission spectra, as plotted in Figure 5f, also showed that this composite transmission line still maintained a big contrast between the transmission parameters S21 and S12, meaning that the energy fluxes unidirectionally propagated along the transmission line in a resistance-free way.

Finally, we verified the topological resistance-free one-way transport property of the transmission lines by comparing the transmission spectra of the three transmission lines with varying levels of complexity. Figure 6a–c show the simulated and measured transmission spectra of the three transmission lines illustrated in Appendix A and Figure 5a,d, respectively. As shown in Figure 6(a1–c1), in simulations, the transmission efficiency of the energy flux in three transmission lines was nearly 100% (the yellow regions ranging from 17.45 to 17.95 GHz). This result reveals that there were no back reflections, mode-mixing problems and impedance-mismatching issues in these topological transmission lines. So, EM waves could propagate forwards in only one direction even when impinging on the intersections between two lines with different wave impedances, and even when repeatedly taking sharp turns. These phenomena were also verified by the measured results illustrated in Figure 6(a2–c2), where the transmission spectra of these three transmission lines exhibited a strong nonreciprocity (about 30~35 dB in contrast) at the frequency range of 17.45~17.95 GHz, although there existed some disturbances in the magnitude. Note that there existed some noises in the experimental results. On the one hand, these noises existed mainly because the source of the vector network analyzer was unstable, and this is very common in microwave experiments [33,34,35,36]. On the other hand, the coupling losses induced by the mode mismatching at the input/output end and the transmission losses that originated from the fabrication errors of the sample also introduced some noises in the experiment results. However, although there existed some noises in the experimental results, the transmission spectra still showed the strong nonreciprocity in the frequency range of 17.45~17.95 GHz, indicating the excellent one-way transport property. Additionally, it should be noted that the perturbation of the external magnetic field strength had almost no influence on the transport behavior of the one-way edge states (see more detailed calculations in the Appendix A). Moreover, Chen et al. also demonstrated that the realization of the one-way edge states in a GPC system has a very strong tolerance to the nonuniform magnetization [40].

## 4. Conclusions and Discussion

In conclusion, we theoretically proposed and experimentally realized a new configuration of PCTIs in a two-dimensional square-hexagon lattice GPC. We found that this PCTI possesses five types of edges and all of them support the one-way propagation of edge states, which can travel around the sharp turns and bypass metallic obstacles in a reflection-free way. We utilized the one-way property of edge states to overcome entirely the issues of back reflection, mode mixing and impedance mismatching at terminates, intersections and discontinuities. We implemented the construction of topological transmission lines of various configurations, some of which even consisted of all five types of edges consecutively connected and thus were very complicated, and demonstrated the topological resistance-free transport in the whole transmission line network. We expect that these results will enrich the geometric and physical infrastructure to implement the PCTIs possessing truly one-way edge states and will bring about novel platforms for developing topology-driven resistance-free photonic devices. Although our work has focused on GPCs, similar ideas can be generalized to other photonic systems, and more broadly to other bosonic platforms, such as acoustics, electrics, mechanics and more.

## Figures and Tables

**Figure 1 nanomaterials-12-03009-f001:**
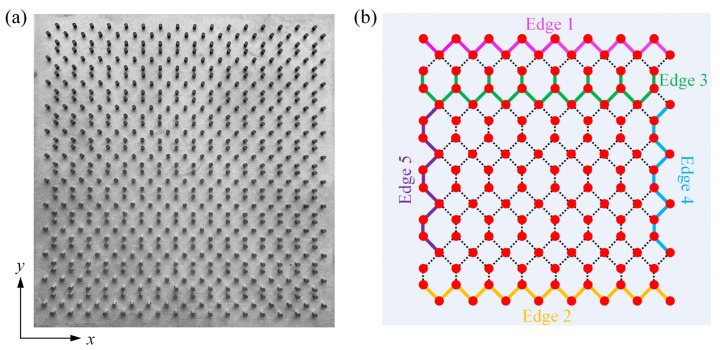
Square-hexagon lattice GPC. (**a**) Photo picture of experimental sample. (**b**) Five types of edges highlighted in five polylines colored in different colors. There were three zigzag-type edges (Edges 1–3) along *x* direction. When the outermost row of YIG cylinders of Edge 1 (outlined with a magenta polyline) is removed, Edge 2 (outlined with a yellow polyline) is formed. When the outermost row of Edge 2 continues to be removed, Edge 3 (outlined with a green polyline) is formed. Two armchair-type edges (Edges 4–5) are directly along *y* direction. When the outermost row of Edge 4 (outlined with blue polyline) is taken away, Edge 5 (marked with purple polyline) emerges.

**Figure 2 nanomaterials-12-03009-f002:**
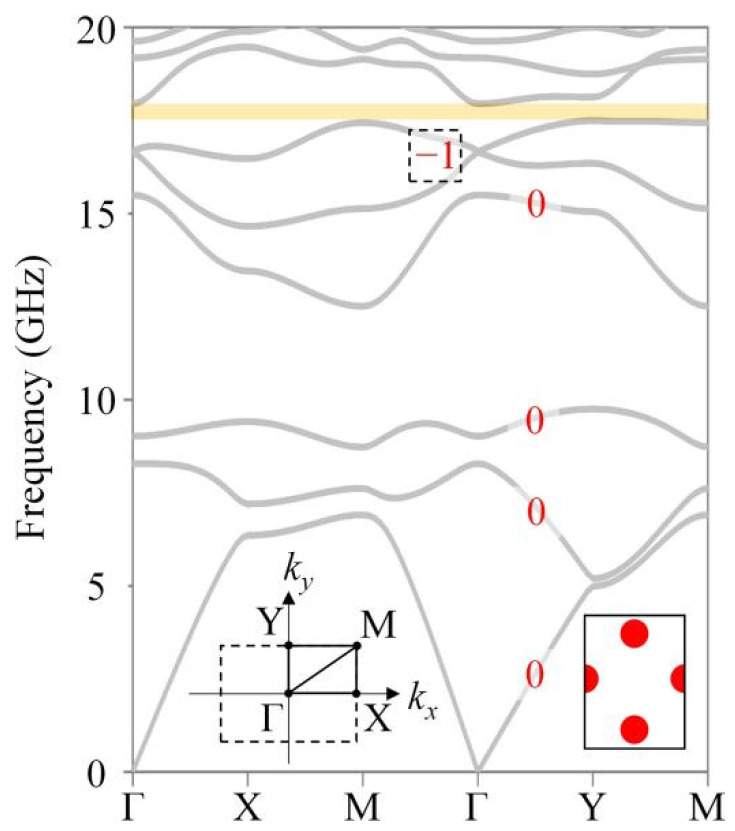
Bulk band of a square-hexagon lattice GPC. There existed a complete bandgap ranging from 17.45 to 17.95 GHz (yellow region) and the Chern number of each band below the second bandgap is marked (red). The Chern number of the first, second, third and fourth bands was zero, while the total Chen number of the intersecting fifth and sixth bands was −1. The lower left inset is the first Brillouin zone of a square-hexagonal lattice. The lower right inset is the unit cell of square-hexagon lattice.

**Figure 3 nanomaterials-12-03009-f003:**
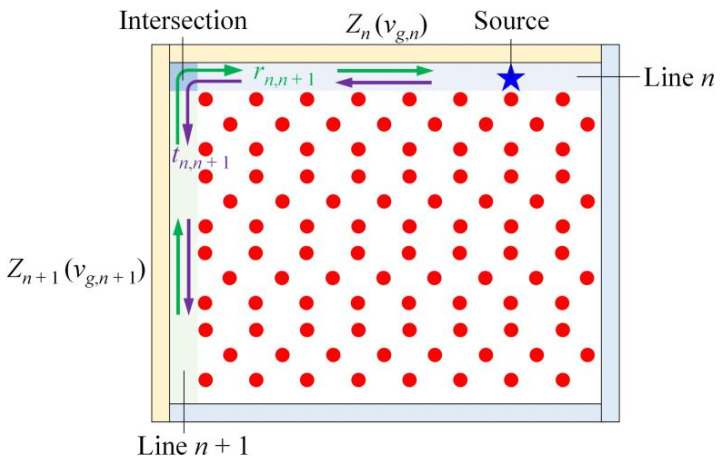
Schematic diagram of matching transmission line network. The upper and left boundaries were set as perfect electric conductors (colored in yellow) to form the transmission lines (Line *n* and Line *n* + 1). The lower and right boundaries were set as the scattering boundary conditions (colored in blue) to avoid the reflection of EM waves. The blue star is the line source, the green and purple arrows at the intersection represent the reflected waves (*r_n_*_,*n*+1_) and transmitted waves (*t_n_*_,*n*+1_), respectively. The impedances (group velocities) of the upper and left transmission lines are *Z_n_* (*v_g,n_*) and *Z_n_*_+1_ (*v_g,n_*_+1_), respectively.

**Figure 4 nanomaterials-12-03009-f004:**
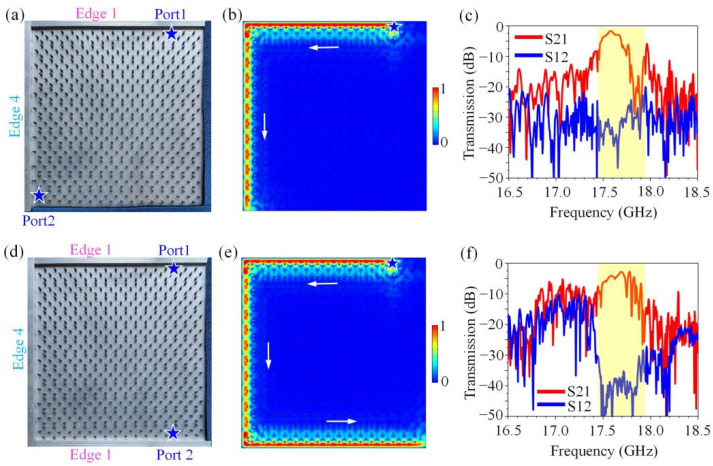
Topological transmission lines composed of Edge 1 and Edge 4. (**a**,**d**) Photos of experimental samples. The metallic strips were placed near the boundaries of GPC to form the transport channels, and the microwave absorbers were utilized to avoid the reflection of EM waves. (**b**,**e**) Simulated electric field distribution excited by a line source. The blue star is the line source and the white arrow indicates the direction of electric field transmission. (**c**,**f**) Measured transmission spectra of topological resistance-free transmission lines. There existed a big contrast (about 35 dB) between parameters S21 and S12 at 17.45~17.95 GHz.

**Figure 5 nanomaterials-12-03009-f005:**
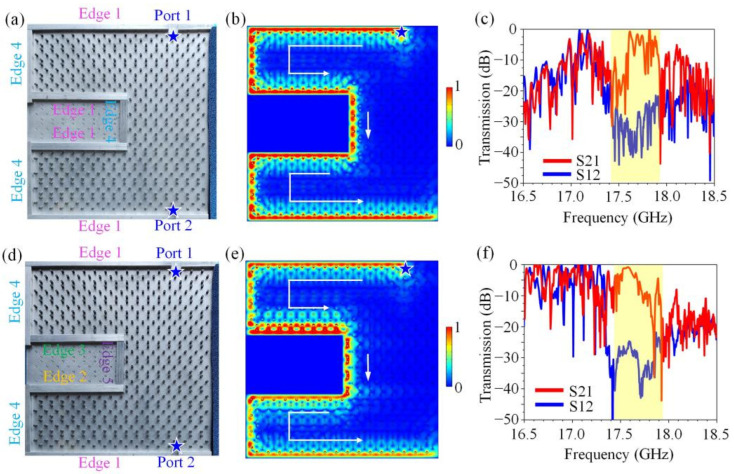
Topological transmission lines composed of Edges 1–5. (**a**,**d**) Photos of experimental samples. The metallic strips were placed near the boundaries of GPC to form the transport channels, and the microwave absorbers were utilized to avoid the reflection of EM waves. (**b**,**e**) Simulated electric field distributions excited by a line source. The blue star is the line source and the white arrow indicates the direction of electric field transmission. (**c**,**f**) Measured transmission spectra of topological resistance-free transmission lines. There existed a big contrast (about 30 dB) between parameters S21 and S12 at 17.45~17.95 GHz.

**Figure 6 nanomaterials-12-03009-f006:**
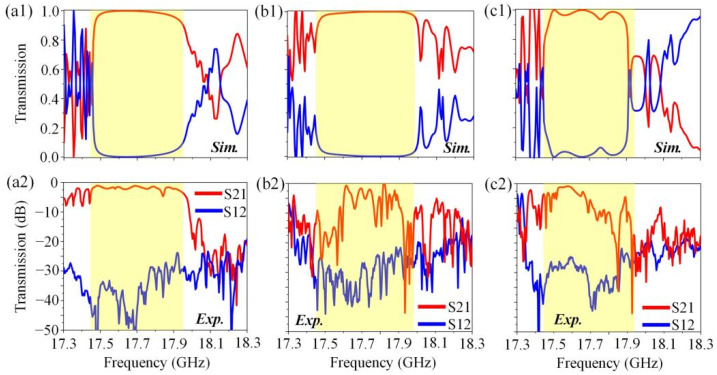
Simulated and measured transmission spectra of three types of topological resistance-free transmission lines. (**a**) Straight transmission line constructed by Edge 1 shown in Appendix A. (**b**) Transmission line with six 90° turns composed of Edge 1 and Edge 4 illustrated in Figure 5a. (**c**) Transmission line with six 90° turns consisted of Edges 1–5 plotted in Figure 5d. (**a1**–**c1**) Simulated and (**a2**–**c2**) measured transmission spectra of three types of topological resistance-free transmission lines.

## Data Availability

The data that support the findings of this study are available from the corresponding authors upon reasonable request.

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
