# Peer review of "Topological Resistance-Free One-Way Transport in a Square-Hexagon Lattice Gyromagnetic Photonic Crystal"

_nanomaterials, 2022, doi:10.3390/nano12173009_

Round 1

Reviewer 1 Report

The paper experimentally and numerically investigated PCTIs using GPCs. The paper is interesting. However, there are some unclear points. Please address the following points.

1.      The novelty is unclear. Is the novelty a square hexagonal lattice gyromagnetic PC? If so, please compare it to other previous studies.

2.      Please describe the experiment set up and the calculation methods more.

3.      The experimental results seem to be very noisy. Please discuss more on the obtained experimental results.

4.      It is kind for readers how to calculate the edge mode and its background.

Reviewer 2 Report

The manuscript “Topological resistance-free one-way transport in a square-hexagon lattice gyromagnetic photonic crystal” by Qin et al. may in principle be interesting, but is severely hold back by some major flaws.

First, while the abstract outlines an external magnetic field, the role of the latter is never discussed or investigated. How are the properties influenced by the strength of the magnetic field? Simulations and further experiments could be useful in this respect. At simulations should be possible.

The descriptions in the figures within the main manuscript are lacking, one need to repeatedly search through the text to get details of the figure, including important information on what is actually shown.

In the SI, is there a reason why the whole “Edge 2” section is written in bold letters?

Furthermore, in my opinion the number of citations is inadequate, and the selection given in the manuscript is not representative of the current state of the art in this field. This needs to be improved, too.

The “transmission parameters S21 and S12”, while probably well known to the authors, are never really explained and may not be immediately clear to the audience of Nanomaterials.

More generally, the language in the main manuscript needs to be improved. There are several issues and sentences that are hard to read. For example:

-) page 1, line 28: in a realistic photonic circuits (either is circuit or delete the “a”)

-) page 1, line 30: The meaning of the sentence starting in this line is not clear to me.

-) throughput the manuscript “impendence” is used (I assume instead of impedance).

-) page 1, line 45: ..edge states emerge with topological protection property.. (plural needs to be used).

-) same on page 2, line 48, 49, 60, 71, 73

-) citation for COMSOL missing.

-) page 3, line 90 “It is known that Chern number..”, a “the” is missing here.

-) page 3, line 108: Sentence starting here is not clear to me, and may not make sense.

-) page 4, line 120: “so if one want to obtain”, instead write “wants to obtain”

-) page 4, line 146: “remain excellent one way property…”, sentence sounds odd.

-) page 5, sentence starting at line 160 makes no sense to me in its current form.

This list by far not complete, and I urge the authors to carefully correct the manuscript as the number of mistakes makes it very hard to read and especially understand.

Currently, I cannot recommend publication of the manuscript in its present form. The manuscript needs major revision at least. Especially the overall language needs to be improved, which is also true to the overall accessibility of the information in the manuscript.

Round 2

Reviewer 1 Report

The authors addressed my commnets well.

Reviewer 2 Report

The authors have made significant and important changes to the manuscript. Only a few typos remain, after which publication in Nanomaterials can be considered.

-) page 3, line 96: "Here, " (add comma)
-) page 3, line 101: "Chern" (misspelled Chern)
-) page 5, line 157: ",illustrated in Fig 4e," (add commas)
-) page 6, line 199: ", as plotted in Fig 5f," (add commas)
-) page 6, line 218: "So, " (add comma)

The investigations of the magnetic field are interesting. Indeed, while the authors state that 100 Gauss is a big perturbation in experiment, it is a very weak perturbation for a molecular system. Using a so to speak "weak" field (weak compared to the Coulomb forces within the molecule!), it is not too surprising to see only minor effects of the field strength, unless the material exhibits strong non-linear effects. The latter effects may be more common in spin polarized systems.
